# Information Bottleneck of Quantum Neural Networks

**Juexiao Wang**
Department of Computer Science
Purdue University
`wang5360@purdue.edu`

**Myeongsu Kim**
Department of Computer Science
Purdue University
`kim2167@purdue.edu`

**Sabre Kais**
Department of Electrical and Computer Engineering
North Carolina State University
`skais@ncsu.edu`

## Abstract

Occam's Razor, as a principle of model selection, favors simpler models that achieve the same empirical error rate to more complex models. Structural risk minimization finds such optimal model with a structural search by minimizing a high probability bound on the true error rate, enjoying the trade-off between minimizing the empirical error rate and the generalization bound. Tishby and Zaslavsky [2015] suggested to quantify such trade-off via the theoretical framework of the Information Bottleneck theory. It claims that Deep Neural Networks learn a compressed bottleneck representation that preserves only the relevant information about the target label $Y$ from the source data $X$. The optimal compression-accuracy trade-off is characterized by the information-bottleneck function, providing a tight achievable lower bound on the representation length.

An open quantum system is equivalent to a noisy neural network that can perform machine learning tasks with non-unitary evolution, leveraging the potential power of entanglement and superposition. Therefore understanding the learnability of a quantum system is crucial in the noisy intermediate-scale quantum (NISQ) era. In this paper, we are generalizing the Information Bottleneck theory to quantum Hilbert space to characterize the learnability of a quantum system with quantum mutual information as the complexity measure of a quantum concept class and the lower bound of the prediction accuracy of a quantum hypothesis. We compared the training dynamics of quantum neural networks with the classical analog in the information plane and, through numerical simulations, demonstrated that the learned quantum representations are more expressive than classical representations.

## 1 Introduction

The Information-Bottleneck (IB) method, introduced in Tishby et al. [2000], considers extracting relevant information about the target signal $Y$ through a correlated observable $X$ with a compressed representation $\hat{X}$. The objective is to find such compressed representation with minimal average description length, characterized by the mutual information $I(X; \hat{X})$, subject to the constraint on the expected distortion as the relevant information $I(\hat{X}; Y)$ about the target signal $Y$. The optimal trade-off between compressing the representation and preserving meaningful information is characterized by the information bottleneck function that separates the achievable and non-achievable region of representations.

38th Second Workshop on Machine Learning with New Compute Paradigms at NeurIPS 2024(MLNCP 2024).

Information-Bottleneck method suggests a learning algorithm known as structural risk minimization, minimizing a high probability bound on the true error rate of the learned hypothesis with respect to a distribution, consisting of empirical error rate and generalization bound.

Mutual information provides a robust measure of the model complexity and the prediction accuracy of a hypothesis. $I(X;\hat{X})$ characterizes the maximal achievable rate of the distinguishable typical sequences with asymptotic zero probability of error which provides an $\epsilon$-cover for the representation space. Relevant information $I(\hat{X};Y)$ provides a lower bound of the prediction accuracy Gilad-Bachrach et al. [2003], Cover and Thomas [2006]. Therefore minimizing the Lagrangian is equivalent to minimizing the high probability bound.

Shwartz-Ziv and Tishby [2017] empirically demonstrated that neural network trained with stochastic gradient descent (SGD) implicitly achieves the Information Bottleneck bound. The optimization dynamics of SGD is a Wiener process with two distinct phases in resonance to the phase transition in thermal annealing dynamics. Empirical risk minimization (ERM) phase minimizes the empirical risks with an increase in $I(\hat{X};Y)$. Representation compression phase has decreasing rate $I(X;\hat{X})$ where the input feature is compressed into a finer grained clustered representation of $X$ such that it can generalize better.

Quantum Neural Networks (QNN) harness the power of entanglement and superposition, to perform machine learning task. A quantum model is a kernel function that maps data into a quantum Hilbert space with dimension exponentially growing with the number of qubits. Quantum channels characterized by unitary evolution transforms the computational basis followed by a projective measurement in such basis defines a linear decision boundary in the quantum feature space. Training a quantum model is equivalent to finding the optimal Positive-Operator Valued Measure (POVM) measurement that minimizes a data-dependent cost function Schuld [2021].

In this paper, we compare the learning dynamics of quantum and classical neural network in the information plane and conclude from the numerical simulation that the learned quantum representations are more expressive than the classical representations.

## 2 Information Bottleneck Theory for Deep Learning

### 2.1 Mutual Information

**Definition 2.1.** (Classical Mutual information) Shannon's mutual information between two random variables $X$ and $Y$ with joint probability distribution $P_{XY}$, is defined as the relative entropy between the joint distribution $P_{XY}$ and the product of its marginal distributions $P_X \otimes P_Y$ .

$$\begin{aligned} I(X;Y) &= D(P_{XY}||P_X \otimes P_Y) \\ &= H(X) + H(Y) - H(X,Y) \end{aligned}$$

The quantum generalization of the mutual information is defined with quantum relative entropy.

**Definition 2.2.** (Quantum Mutual Information) Quantum mutual information is defined as the quantum relative entropy between the joint state $\rho_{XY}$ and the product state $\rho_X \otimes \rho_Y$ of individual subsystems.

$$\begin{aligned} I(X;Y)_\rho &= S(\rho_{XY}||\rho_X \otimes \rho_Y) \\ &= H(X)_\rho + H(Y)_\rho - H(X,Y)_\rho \end{aligned}$$

### 2.2 Information Bottleneck of classical DNNs

Information Bottleneck method introduced in Tishby et al. [2000] finds the optimal code for the source signal that preserves the relevant information about a target signal with minimal codeword length. Let $X$ denote the signal space with a probability distribution $P_X$ and let $\hat{X}$ denote its quantized codebook. For every signal element $x \in X$, it is mapped to a codeword $\hat{x} \in \hat{X}$ via a stochastic map $P_{\hat{X}|X}$.

The optimal representation $\hat{X}$ is an approximate minimal sufficient statistics of $X$ with respect to $Y$ that trades off between compressing the representation of $X$ and preserving the meaningful information about $Y$. Tishby et al. [2000] characterizes this trade-off through the information bottleneck function $R^I(I_Y)$, as the minimal achievable rate under a given constraint on the expected distortion measured by $I(\hat{X}; Y)$ Gilad-Bachrach et al. [2003].

$$R^I(I_Y) = \min_{P_{\hat{X}|X}:I(\hat{X};Y)\geq I_Y} I(X; \hat{X}) \tag{1}$$

Finding the information-bottleneck function is a variational problem that can be solved by minimizing the Lagrangian functional

$$L[\,P_{\hat{X}|X}\,] = I(\hat{X}; X) - \beta I(\hat{X}; Y) \tag{2}$$

The positive Lagrange multiplier $\beta$ controls the tradeoff between the complexity of the representation $I(X; \hat{X})$ and the relevant information $I(\hat{X}; Y)$ to recover $Y$. By varying the parameter $\beta$ one can explore the trade-off between the preserved meaningful information and compression at various resolutions Tishby et al. [2000].

When applying the information bottleneck principle to DNNs, the layered structure of the network generates a successive Markov chain of intermediate representations, which together form the (approximate) sufficient statistics.

### 2.3 Information Bottleneck of quantum DNNs

Catli and Wiebe [2022] introduces a quantum generalization of the information bottleneck theory by replacing the KL-divergence with the quantum relative entropy in the definition of the mutual information.

**Definition 2.3.** (Quantum Relative Entropy) Let $\rho_A$ and $\rho_B$ be density operators acting on $C^{2^n}$ for integer $n$, the quantum relative entropy is defined to be

$$S(\rho_A||\rho_B) = \text{Tr}(\rho_A log(\rho_A)) - \text{Tr}(\rho_A log(\rho_B)) \tag{3}$$

**Definition 2.4.** (Quantum Information Bottleneck Objective Function) The quantum information bottleneck objective function for a distribution $\rho_{\hat{X}XY}$ and a quantum channel $\phi$ is denoted $L[\phi]$ which is defined to be

$$L[\phi] = I(\hat{X}; X)_\rho - \beta I(\hat{X}; Y)_\rho = S(\rho_{X\hat{X}}||\rho_X \otimes \rho_{\hat{X}}) - \beta S(\rho_{\hat{X}Y}||\rho_{\hat{X}} \otimes \rho_Y) \tag{4}$$

To understand how the relevant information flows through a quantum network, Catli and Wiebe [2022] introduces a quantum embedding of classical data $x \mapsto \rho(x)$ by encoding the classical joint probability distribution $P_{XY}$ in a bipartite quantum state. Consider a training set $S = \{(x_i, y_i)\}_{i=1}^n$, the equivalent quantum embeddings is characterized by the density operator of the ensemble

$$\mathcal{E} \equiv \{\hat{P}_{XY}(x_i, y_i), \rho(x_i) \otimes \rho(y_i)\}_{i=1}^n \tag{5}$$

$$\rho_{XY} = \sum_{x,y} \hat{P}_{XY}(x,y) |x\rangle \langle x| \otimes |y\rangle \langle y| \tag{6}$$

$$\rho_{\hat{X}Y} = \sum_{\hat{x},y} \hat{P}_{\hat{X}Y}(\hat{x},y) |\hat{x}\rangle \langle \hat{x}| \otimes |y\rangle \langle y| \tag{7}$$

where $\hat{P}_{XY}(x,y)$ is an empirical estimate of the joint distribution $P_{XY}$. The density operators $\rho_X, \rho_{\hat{X}}$ and $\rho_Y$ can then be found by taking partial traces of the bipartite states.

$$\rho_X = \text{Tr}_Y \rho_{XY} = \sum_x \hat{P}_X(x) |x\rangle \langle x| \tag{8}$$

$$\rho_{\hat{X}} = \text{Tr}_Y \rho_{\hat{X}Y} = \sum_x \hat{P}_{\hat{X}}(\hat{x}) |\hat{x}\rangle \langle \hat{x}| \tag{9}$$

$$\rho_Y = \text{Tr}_X \rho_{XY} = \sum_y \hat{P}_Y(y) |y\rangle \langle y| \tag{10}$$

The quantum embeddings above are considered as *classical* states when the states $\{|x\rangle\}_{i=1}^{n}$, $\{|\hat{x}\rangle\}_{i=1}^{n}$ and $\{|y\rangle\}_{i=1}^{n}$ form an orthonormal basis, reflecting that the states of two quantum subsystems $\rho_X, \rho_Y$ are only classically correlated Grimsmo and Still [2016].

In the following numerical simulation, we consider the quantum embeddings $\{\hat{P}_X(x_i), \rho(x_i)\}_{i=1}^{n}$ to be orthogonal to each other so that the classical mutual information of a quantum channel can be extracted to have a fair comparison with the classical channel in the output representation complexity.

# 3 Mutual Information Estimation

Mutual information is a nonlinear functional of a joint probability distribution $P_{XY}$ over the product measurable space $\mathcal{X} \times \mathcal{Y}$ and it is invariant under invertible transformations. In a deterministic deep neural network, the true mutual information $I(X; \hat{X}_l)$ between the input $X$ and the output of one of its internal layer $\hat{X}_l$ is provably a constant if $X$ is a discrete random variable or infinite if $X$ is a continuous random variable Goldfeld et al. [2019]. Therefore it is meaningless to track information flow in a deterministic network where $I(X; \hat{X}_l)$ is independent of the system parameters and the learned representations.

## 3.1 Classical mutual information estimation

Goldfeld et al. [2019] propose a stochastic DNN framework by injecting isotropic Gaussian noise to the output of each of the DNN's hidden layer neurons. The stochastic map makes $I(X; \hat{X}_l)$ evolves with the learned representations. Goldfeld et al. [2019] construct a provably accurate estimator of differential entropy under Gaussian convolution by collecting monte carlo samples. Estimating the mutual information in a classical noisy DNNs is reduced to estimating the differential entropy of the representation distribution under Gaussian convolution. We apply the non-parametric Kernel Density estimator and k-NN based estimator in Saxe et al. [2018] for mutual information estimation in the training of classical neural networks.

## 3.2 Quantum mutual information estimation

Analogous to the classical noisy DNNs, we inject quantum noise to each qubit in QNNs, in particular we employ the amplitude damping channel as a layer of QNNs. Since the classical mutual information of the bipartite state $\rho_{\hat{X}Y} = \sum_{\hat{x},y} \hat{P}_{\hat{X}Y}(\hat{x}, y) |\hat{x}\rangle \langle\hat{x}| \otimes |y\rangle \langle y|$ characterizes the $\epsilon$-cover of the quantum concept class, we are extracting the classical mutual information of the quantum channel. By estimating the von-Neumann entropy of an ensemble of orthogonal states, we are essentially extracting the classical correlation between input and output states of a quantum channel characterized by $I(X; Y)$.

We thank the reviewer for the insightful comments, to estimate the classical mutual information of a quantum state under non-unitary evolution, in high dimensional space, non-unitary transformation preserves the orthogonality between states with high probability so the von-Neumann entropy of the bipartite state is equal to the classical Shannon entropy wih high probability.

# 4 Numerical simulation

We compared the learning dynamics of classical neural networks and quantum neural networks on a constructed dataset in the information plane.

## 4.1 Dataset

The dataset is constructed to be a set of standard normal basis vectors $S = \{(x_i, y_i)\}_{i=1}^{N}, x_i^T x_j = 0, \forall i \neq j$ and we apply the amplitude encoding to map the classical training example $x_i$ to a quantum state $|x_i\rangle = \sum_{d=1}^{N} x_i^d |d\rangle$.

Moreover, the training samples are linearly separable with a full rank feature matrix $X$ thus there exists a unique linear separator that is consistent with the training samples $w = X^{-1}y$ which provides a good sanity check on the training accuracy achieved by the two models.

## 4.2 Neural Network Architecture

To have a fair comparison between the expressiveness of quantum neural networks and classical neural networks, we employ a classical neural network architecture with identity activation function with the same number of parameters as the quantum neural networks. In resonance to Shwartz-Ziv and Tishby [2017], we have the Quantum Neural Network composed of a sequence of 6 non-unitary operators composed of rotation gates on each qubits and pairwise CNOT gates.

## 4.3 Classical vs Quantum information plane

We compared the classical training dynamics with quantum training dynamics in the information plane. The training trajectories are shown in Fig.1.

### 4.3.1 Sample complexity determined by $I(X; \hat{X})$

For large typical input patterns Shwartz-Ziv and Tishby [2017], $2^{I(X;\hat{X})}$ provides an $\epsilon$-cover of the hypothesis class. The sample complexity of agnostic PAC learning a concept class is characterized by $\mathcal{M}_{ERM}^{AG} = O(\frac{2}{\epsilon^2}(2^{I(X;\hat{X})} + \ln(\frac{2}{\delta})))$.

At the end of the training with dataset consisting of 64 standard basis vectors, the mutual information between quantum representations and the input is shown to be greater than that between classical representations and the input $I(X; \hat{X}_Q) \approx 4.6 > I(X; \hat{X}_C) \approx 1$. The result can be interpreted as that classical representations are clustered into two clusters $I(X; \hat{X}_C) = H(\hat{X}_C) - H(\hat{X}_C|X) \approx H(\hat{X}_C) = 1$ while the quantum representations have more patterns due to the richness of the quantum Hilbert space, which implies that the sample complexity to achieve the same generalization guarantee for quantum neural networks is higher than that for the classical neural networks.

We conjecture that the higher degree of the expressiveness of quantum representations is a result of the richness of the quantum Hilbert space, which is a complex vector space with more elements than the real vector space and we can potentially prove it using the chain rule of mutual information under the assumption that the input distribution $P_X$ and the noise model $P_{\hat{X}|X}$ are the same and the real component $\hat{X}_R$ and imaginary component $\hat{X}_I$ of the representation are distributed independently and identically by decomposing the mutual information into $I(X; \hat{X}_Q) = I(X; \hat{X}_R) + I(X; \hat{X}_I) \geq I(X; \hat{X}_R) = I(X; \hat{X}_C)$ Xie et al. [2014].

### 4.3.2 Test accuracy determined by $I(\hat{X}; Y)$

For large typical input patterns Shwartz-Ziv and Tishby [2017], $I(\hat{X}; Y)$ provides a lower bound on the testing accuracy. At the end of the training with 64 standard normal basis vectors, the mutual information between quantum representation and the target label is less than that between classical representation and the target label $I(\hat{X}_Q; Y) \approx 0.9 > I(\hat{X}_C; Y) \approx 0.7$, which might imply that the quantum representations can provide better generalizations than the classical representations.

### 4.3.3 Loss landscape

We also observe that there exists a flatten area in the quantum information plane, as shown in Fig. (1), where there is high degeneracy in $I(Y; \hat{X})$ which might be an indication of barren plateau in the quantum loss landscape Mcclean et al. [2018].

## 5 Conclusion

In order to study the fundamental question in learning theory, the quantification of the basic tradeoff between the complexity of a model and its predictive accuracy, Tishby and coworkers introduce the

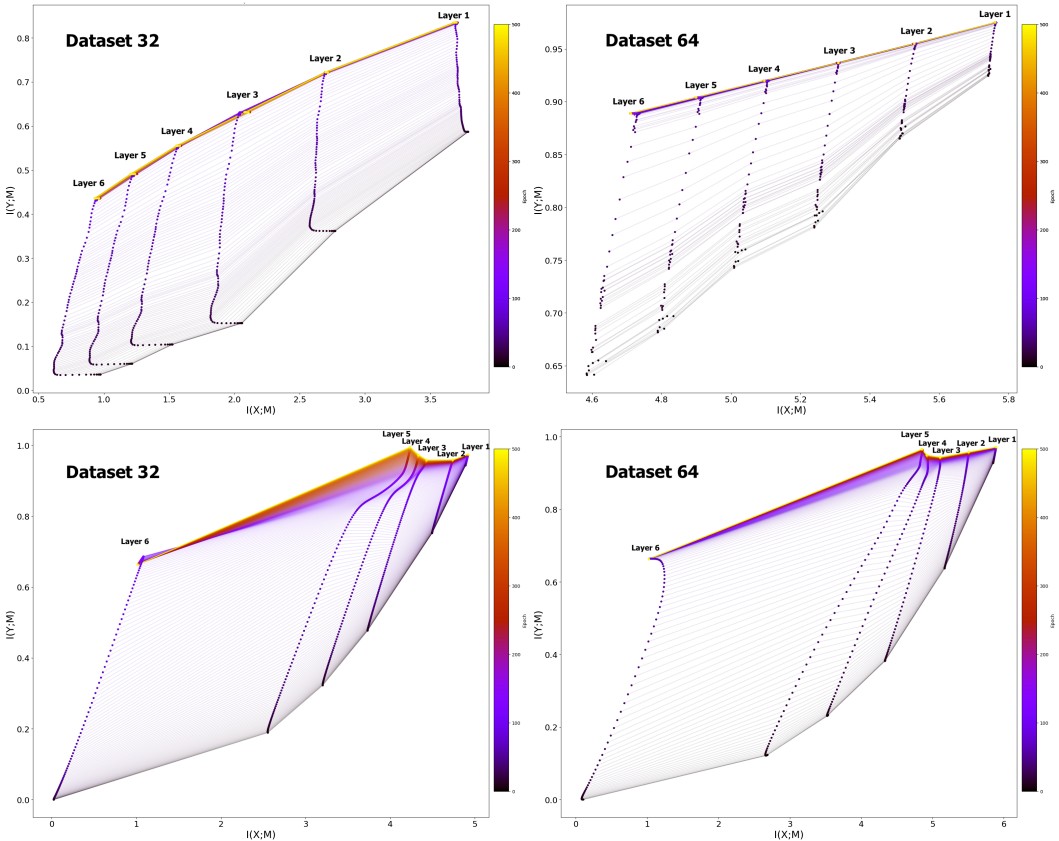

Figure 1: Information Plane of the training dynamics of a Quantum Neural Network (upper) and Classical Neural Network (lower). The x-axis ($I(X;M)$) shows the mutual information between the input X and the compressed representation M, while the y-axis ($I(Y;M)$) represents the mutual information between the target Y and the representation M, illustrating how much relevant information is retained for prediction.

concept of "Information Bottleneck". Thus, Shannon's mutual information allows us to quantify the tradeoff and measure both the model's complexity and its prediction accuracy. Their work focuses on classical machine learning by studying the nature of this complexity-accuracy tradeoff and discuss some of its theoretical properties. Furthermore, they present relations to classical information theoretic problems, such as rate-distortion theory, cost-capacity tradeoff and source coding with side informationTishby and Zaslavsky [2015].

In this paper, we compared the training dynamics quantum neural network and classical neural network in the information plane. We demonstrated through numerical simulations that quantum neural network is a more expressive concept class which can fit dataset with more complicated structures thus decreasing the approximation error, while taking more samples to achieve the same generalization guarantee thus increasing the estimation error. For future work, further simulation results might be needed to provide more convincing conclusions and exact bounds. The implementation and related resources for this work are available at `https://github.com/kim2167/Information_Bottleneck_QNN`.

# 6   Acknowledgement

We would like to thank Dr. Manas Sajjan for fruitful discussions. J.W. thanks the members from Kais group for helpful discussions and the precious suggestions and feedback received from the discussions with several professors throughout the project.

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
