# OpenReview forum: "Information Bottleneck of Quantum Neural Networks"
_NeurIPS.cc/2024/Workshop/MLNCP — MLNCP Poster_

### Official Review · Reviewer_LJ7n · 2024-09-18

**Rating:** 3
**Confidence:** 4

**Review:**

This paper attempts to study the expressiveness of quantum neural networks via information bottleneck, and compares it with classical neural networks. The authors conducted numerical experiments to calculate information quantities on synthetic data to illustrate their claims that quantum neural networks are more expressive than classical ones.  Despite this idea being interesting, I find that the setting considered in this paper is somewhat flawed. Detailed comments are listed below.

1. One major concern is that the representation produced by quantum neural networks (i.e., those density matrices) are quantum objects that will be collapsed when measured. This is different from classical representations. But the authors seem to neglect this fact and use classical information quantity directly to the density matrices. This procedure effectively treats the density matrix as equivalent to its classical representation, which is wrong because to get classical representation from quantum objects, you need tomography that requires exponentially many copies to perform. To put it in another way, from Holevo's theorem, one can only extract $n$ classical bits of information from an $n$-qubit mixed state, but a classical description of an $n$-qubit mixed state contains $\exp(n)$ bits of classical information. The author's procedure essentially neglected this gigantic gap.

2. The argument in 4.1 that the quantum entropies can be replaced by classical ones is not correct. It's correct that for diagonal matrices, the quantum entropy is the same as the classical one. But when dealing with relative entropy that involves two states, the relative rotation between the two sets of eigenvectors matters. So the quantum relative entropy is not the same as classical ones. The same is true for mutual information.

Other minor comments:

3. Line 45, the authors should unify their notation in using $\tilde{X}$ or $T$ as the intermediate representation.

4. Line 99, Def. 2.3 should be placed before Def. 2.2.

5. Eq. (7). the notation is very confusing. I suppose $\hat{X}$ should be $\tilde{X}$? The state $\rho_{\tilde{X}Y}$ is never defined, and it's not clear how $\ket{\hat{x}}$ is related to $\ket{x}$. Also I'd like to remind the authors that when measuring the intermediate representation $\tilde{X}$, there're many POVM measurements one can choose from, which do not necessarily measure in the basis $\ket{\hat{x}}$ that diagonalizes $\rho_{\hat{X}}$. So it's not correct to directly treat it as Eq. (7).

6. Line 154, it's not clear to me why the activation function of classical neural nets is set to identity. It's not a fair comparison since in quantum neural nets, the data-encoding and measurement can introduce non-linearity.

---

### Official Review · Reviewer_LRgh · 2024-09-21
**Interesting idea, suggest more systematic numerical studies**

**Rating:** 7
**Confidence:** 4

**Review:**

The authors generalize the classical information bottleneck function, by instead considering the quantum mutual information between quantum states. They then numerically investigate how this function differs in learning tasks where the data is encoded quantumly rather than classically. The authors then speculate that the different behavior of this function in the quantum and classical settings explains the different learning and generalization performances of the quantum and classical models.

Though this is obviously early work, it is very intriguing and a novel way to understand the distinctions between quantum and classical networks. In the full version of this result I'd be interested in seeing how this quantity scales in, for instance, quantum networks known to suffer from barren plateaus and those which do not (see, e.g., [arXiv:2312.09121](https://arxiv.org/abs/2312.09121)); those that suffer from poor local minima and those that do not (see, e.g., [arXiv:2408.11901](https://arxiv.org/abs/2408.11901)); and so on.

---

### Official Review · Reviewer_MFLX · 2024-10-08

**Rating:** 4
**Confidence:** 3

**Review:**

**Summary:** \
The article applies the seminal concept of the information bottleneck to compare the learning dynamics of quantum and classical neural networks using a simple toy dataset of standard basis vectors. Numerical simulations demonstrate that the representations learned by quantum neural networks (QNNs) are more expressive than their classical counterparts.
\
\
**Main Review:**
\
**Writing:** The overall presentation and writing style could be improved, as there are several typographical and grammatical errors throughout the text. While I did not exhaustively identify every mistake, some examples include "simplifies" instead of "simplified" and "achived" instead of "achieved." It is recommended that the authors run the manuscript through automated grammar-checking tools (e.g., Grammarly) or a language model to enhance the flow and clarity of the text.
\
\
**Only One source of quantum noise:** To investigate the information flow in noisy QNNs, the authors have only utilized the amplitude damping channel as the source of quantum noise. However, this alone does not adequately represent the behavior of QNNs on a real quantum device. While acknowledging that access to quantum hardware can be limited due to cost, it is recommended that the authors incorporate additional noise models such as depolarizing noise, thermal relaxation, readout noise, and shot noise. This would provide a more comprehensive and realistic assessment of QNN performance, ensuring that the results obtained hold under conditions closer to those of an actual quantum computer.
\
\
**No data-reuploading**: The absence of data re-uploading in the QNN is acceptable; however, given the authors' objective of demonstrating that QNNs can learn better representations than classical deep neural networks (DNNs), including data re-uploading could enhance the expressivity of the QNN and strengthen their claim.
\
\
**Code:** Another area of improvement is the lack of anonymized source code, which hinders the reproducibility and further development of the work. Providing accessible code would significantly contribute to the transparency and credibility of the results.
\
\
**Miscellaneous details missing:** The article omits several critical technical details, such as the dimensionality of each data point, the number of parameters used in both QNN and DNN models, the number of qubits, layers in the parametrized quantum circuit, and specifics of the optimizer and learning rate. These details are essential for reproducing and validating the work, and their inclusion would greatly enhance the clarity and usefulness of the article.
 \
\
**Results limited to one dataset:** The analysis appears limited to the specific dataset and the particular QNN and DNN architectures chosen in this work. It would be valuable to extend the scope of the study and explore whether the findings can be generalized beyond this specific configuration. For instance, a follow-up analysis could involve testing a toy dataset that is not linearly separable while keeping all other variables constant to establish the robustness of the claims made.
\
\
**Loss landscape and Information plane:** The authors suggest that the observed flatness of the information plane in the QNN might indicate the presence of a barren plateau. However, such a bold claim requires theoretical substantiation. Unless the authors can provide rigorous proof to support this assertion, it is advised to refrain from making such speculative statements.

---

### Decision · Program_Chairs · 2024-10-10

Accept (Poster)